# The Protective Effect of Simvastatin on the Systolic Function of the Heart in the Model of Acute Ischemia and Reperfusion Is Due to Inhibition of the RhoA Pathway and Independent of Reduction of MMP-2 Activity

**DOI:** 10.3390/biom12091291

**Published:** 2022-09-13

**Authors:** Monika Skrzypiec-Spring, Agnieszka Sapa-Wojciechowska, Alina Rak-Pasikowska, Maciej Kaczorowski, Iwona Bil-Lula, Agnieszka Hałoń, Adam Szeląg

**Affiliations:** 1Department of Pharmacology, Wrocław Medical University, 50-345 Wrocław, Poland; 2Department of Clinical Chemistry, Wrocław Medical University, 50-556 Wrocław, Poland; 3Department of Clinical and Experimental Pathology, Wrocław Medical University, 50-556 Wrocław, Poland

**Keywords:** simvastatin, matrix metalloproteinases, RhoA, ischemia/reperfusion

## Abstract

The present study investigated whether Rho-associated protein kinase (RhoA/ROCK) signaling pathway inhibitor simvastatin inhibits matrix metalloproteinase 2 (MMP-2) activity in a rat ischemia-reperfusion injury (I/Ri) model by inhibiting the RhoA/ROCK pathway and reducing *MMP-2* mRNA levels. Isolated rat hearts were subjected to aerobic perfusion or I/Ri control. The effect of simvastatin was assessed in hearts subjected to I/Ri. We determined cardiac mechanical function, the content of RhoA, phosphorylated myosin light chain subunit 1 (phospho-MYL9), troponin I, MMP-2, and *MMP-2* mRNA in the heart homogenates, as well as MMP-2 activity in heart tissue. We showed that treatment with simvastatin caused improvement in the contractile function of the heart subjected to I/Ri which was accompanied by a decrease of MMP-2 activity in heart tissue along with inhibition of RhoA pathway, expressed in a reduction in both RhoA and its downstream product—phosphorylated myosin light chain (phospho-MYL9) in hearts treated with simvastatin. MMP-2 inactivation is not due to inhibition of *MMP-2* m-RNA synthesis caused by inhibition of RhoA/ROCK pathway and is due, at least in part, to the direct drug action. The protective effect of simvastatin on systolic function in the acute ischemia-reperfusion model does not appear to be related to reduced MMP-2 activation, but other mechanisms related with the inhibition RhoA/ROCK pathway.

## 1. Introduction

Despite advances in the treatment of myocardial infarction, the disease continues to lead to the development of heart failure. Therefore, the prevention of damage to the mechanical function of the heart muscle appears to be a promising treatment strategy for this disease.

One therapeutic target in preventing damage to the mechanical function of the heart is metalloproteinases (MMPs). While it is well known that MMPs contribute to the degradation of extracellular matrix substrates, they have also been shown to be localized intracellularly and contribute to acute myocardial injury in the ischemia-reperfusion model. MMPs are activated as a result of oxidative stress induced by ischemia-reperfusion injury and cause the cleavage of troponin I, myosin-1 light chain, α-actinin, and titin [1].

We have previously shown that inhibition of metalloproteinase 2 activity in the ischemia and reperfusion model by okadaic acid and carvedilol prevents mechanical damage to the myocardium by inhibiting troponin I degradation [2,3]. We have also shown that inhibition of metalloproteinase 2 by carvedilol reduces troponin I degradation, myofilaments destruction, and mechanical damage to the heart in an autoimmune myocarditis model [4].

It has been shown that MMP-2 activation may be inhibited by statins at the transcriptional and posttranscriptional levels [5,6,7,8,9,10,11]. The theory that statins may induce epigenetic control of MMP was also tested [12]. We previously demonstrated a reduction in mechanical cardiac dysfunction, damage to myofilaments, and a reduction in troponin I degradation by simvastatin in a model of myocarditis, which exerted this effect by inhibiting MMP-9 activity [13]. 

Simvastatin as well as lovastatin and atorvastatin have been shown to inhibit the activation of MMP-2 and MMP-9 by down-regulating the phosphorylation of the myosin phosphatase target subunit 1 (MYPT1) and myosin light chain (MLC), which are downstream substrates of Ras homolog family member A/Rho-associated coiled-coil containing kinases (RhoA/ROCK) signaling pathway in astrocytes of the human optic nerve head [14]. It has also been shown that simvastatin reduces the secretion of MMP-9 from human saphenous vein smooth muscle cells by inhibiting the RhoA/ROCK pathway and reducing the level of *MMP-9* mRNA [15].

ROCKs, or Rho kinases, belong to the serine/threonine protein kinases A, G and C [16]. There are two isoforms of ROCKs: ROCK1 and ROCK2, encoded by two different genes [17]. Excessive activation of ROCKs is involved in various cardiovascular pathological processes, such as cardiac hypertrophy, apoptosis, systemic and pulmonary hypertension [18]. 

The Rho kinase, which is closely related to cell apoptosis, is ROCK1 [19]. ROCK1 preferentially locates in actomyosin filament bundles and, upon activation, phosphorylates myosin light chain, which causes apoptotic membrane blebbing [20]. The pro-apoptotic role of ROCK1 has been demonstrated in the myocardium [21,22]. Clinical studies have also shown that increased ROCK activity is a predictor of worsening prognosis in patients with myocardial infarction [23]. Moreover, ROCK inhibitors have been shown to reduce apoptosis of myocardial cells and improve the impaired mechanical function of the heart in an ischemia-reperfusion model or induced myocardial infarction in rats [24,25].

This study investigated whether another RhoA/ROCK signaling pathway inhibitor, simvastatin, decreased MMP-2 activity in a rat model of ischemia-reperfusion injury by inhibiting the RhoA/ROCK pathway and reducing *MMP-2* mRNA levels.

## 2. Materials and Methods

### 2.1. Animals for Acute IR-Injury

The isolated perfusion of rat hearts was performed on 20 male Wistar rats, weighing 250–350 g. The animals were purchased from the Laboratory Animal Center of Wroclaw Medical University.

After pretreatment with simvastatin (20 mg/kg) or vehicle (intraperitoneally, 18 ± 2 and 3 ± 1 h before anesthesia) rats were anesthetized with intra-peritoneal administration of thiopental (75 mg/kg). Hearts were then excised, and following rapid cannulation of the aorta, retrograde coronary perfusion with Krebs–Henseleit buffer was performed. The Krebs–Henseleit buffer was composed as follows: NaCl 118 mmol/L, KCl 4.7 mmol/L, KH_2_PO_4_ 1.2 mmol/L, MgSO_4_ 1.2 mmol/L, CaCl_2_ 3.0 mmol/L, NaHCO_3_ 25 mmol/L, glucose 11 mmol/L, and EDTA 0.5 mmol/L and gassed with 95% O_2_/5% CO_2_, pH 7.4. Perfusion was performed at the constant pressure (60 mmHg) at the temperature of 37 °C. Left ventricular pressure was monitored with the use of a fluid-filled latex balloon inserted into the left ventricle. Left ventricular developed pressure (LVDP) was calculated as the difference between systolic and diastolic pressures. Additionally, flow rate and heart rate were recorded. Cardiac mechanical function was expressed as the rate-pressure product (RPP) calculated as the product of the spontaneous heart rate and LVDP.

After 25 min of aerobic perfusion, hearts were subjected to 20-min-long global, no-flow ischemia. Then the flow was restored, and the hearts were perfused aerobically for 30 min. In the control group hearts were perfused aerobically for 75 min. At the end of the protocol hearts were frozen clamped in liquid nitrogen and stored at −80 °C for further analysis.

### 2.2. Preparation of the Tissue Samples for RNA Extraction and Biochemical Analysis

Hearts were immersed in liquid nitrogen and grinded using porcelain mortar and pestle. A portion of 50 mg of the tissue powder was transferred to 1 mL of PureZol RNA isolation reagent (BioRad, Hercules, CA, USA) and homogenized by Pellet Pestle^®^ Motor (Kimble Kontes, St. Louis, MO, USA). Samples were frozen at −80 °C until further processing. Another portion of 50 mg of tissue powder was mixed with homogenization buffer containing 50 mmol/L Tris-HCl (pH 7.4), 150 mmol/L NaCl, 0.1% Triton X-100, and Protease Inhibitors Cocktail Set III (Sigma-Aldrich, St. Louis, MO, USA) and homogenized on ice by Pellet Pestle^®^ Motor (Kimble Kontes, St. Louis, MO, USA) to prepare 20% tissue homogenates (*w*:*v*). Homogenization buffer for preparation of samples for phospho-MYL9 Western blot determination additionally contained 1% of Phosphatase Inhibitor Cocktail 2 and 1% of Phosphatase Inhibitor Cocktail 3 (Sigma-Aldrich, St. Louis, MO, USA). Samples were centrifuged at 4 °C and the protein content in the supernatant was measured using Bradford Protein Assay (Bio-Rad, Hercules, CA, USA) using Bovine Serum Albumin (BSA, heat shock fraction, Sigma-Aldrich, St. Louis, MO, USA) as a protein standard. Supernatants were stored at −80 °C until biochemical analysis.

### 2.3. RNA Extraction and Quantitative Polymerase Chain Reaction

Samples frozen in PureZol reagent were thawed and subjected to the RNA extraction procedure according to the manufacturer’s instruction. Briefly, the chloroform (Stanlab, Lublin, Poland) extraction was performed, followed by precipitation of RNA from the water phase with isopropanol (Chempur, Piekary Slaskie, Poland) and washing with 75% ethanol (Chempur, Piekary Slaskie, Poland). Dried RNA pellets were dissolved in 50 μL of ultra-pure DEPC-treated water (Ambion, Waltham, MA, USA). RNA concentration and purity were assessed using NanoDrop Lite Spectrophotometer (Thermo Fisher Scientific, Waltham, MA, USA). Reverse transcription (RT) was performed on 500 ng of each RNA preparation using iScript™ cDNA Synthesis Kit (BioRad, Hercules, CA, USA) following the manufacturer’s instruction. Real-time PCR was performed on 100 ng of each cDNA template in duplicate for both genes and iTaq Universal SYBR^®^ Green Supermix (BioRad, Hercules, CA, USA) according to manufacturer’s protocol. RT and real-time PCR were both performed using CFX96 Touch Real-Time PCR Detection System (BioRad, Hercules, CA, USA). Primers sequences for *MMP-2* and glyceraldehyde 3-phosphate dehydrogenase (*GAPDH*) were designed as follows: rat MMP-2 F: 5′ AGCAAGTAGACGCTGCCTTT 3′, R: 5′ CAGCACCTTTCTTT-GGGCAC 3′; rat GAPDH F: 5′ AG-TGCCAGCCTCGTCTCATA 3′, R: 5′ GATGGTGATGGGTTTCCCGT 3′. *GAPDH* served as a housekeeping gene for normalization of *MMP-2* gene expression, and the relative fold of change in *MMP-2* gene expression was calculated on the basis of delta-delta Ct formula.

### 2.4. Measurement of MMP-2 by Gelatin Zymography

The activity of gelatinases was assessed in heart tissue extracts by gelatin zymography. Samples containing 20 μg of protein were separated on 7.5% polyacrylamide gels copolymerized with gelatin from porcine skin (Sigma-Aldrich, St. Louis, MO, USA) at a concentration of 2 mg/mL of gel, containing 0.1% SDS. After electrophoresis gels were washed three times for 20 min in 2.5% Triton X-100 to restore gelatinases activity and then placed in zymogram incubation buffer (50 mmol/L Tris-HCl, 10 mmol/L CaCl_2_, 200 mmol/L NaCl, and 0.05% NaN_3_) at 37 °C for 18 h. Then gels were stained in 0.5% Coomassie Brilliant Blue R-250 in 30% methanol/10% acetic acid for 2 h, and destained in 30% methanol/10% acetic acid until the white bands on dark blue background were clearly visible. Gels were scanned using GS-800 Calibrated Densitometer with Quantity One v.4.6.9 software (BioRad, Hercules, CA, USA) and the relative MMPs activity was determined and expressed in arbitrary units (AU) calculated on the basis of MMP-2 activity in human capillary blood standards according to Makowski and Ramsby which was separated in each gel [26]. 

### 2.5. In Vitro MMP-2 Inhibition Experiment

This experiment was performed to assess the direct influence of simvastatin on gelatinolytic activity of MMPs separated during electrophoresis on polyacrylamide gelatin-containing gels by incubation of zymograms in the buffer containing the drug. Simvastatin was dissolved in dimethyl sulfoxide (DMSO, Sigma-Aldrich, St. Louis, MO, USA). Zymography of heart samples from groups not pre-treated with simvastatin and two kinds of standards of MMPs activity was performed as described above, but at the step of zymogram development gels were placed either: (1). in buffer containing DMSO (300 μL/100 mL of buffer) serving as a control; or (2). buffer with addition of the same volume of simvastatin stock solutions to achieve final concentration 3 μmol/L; or (3). buffer with the addition of the same volume of a proper stock solution of simvastatin to achieve final concentration 30 μmol/L in zymogram incubation buffer. After 18 h of incubation gels were scanned and the results were calculated on the basis of AUC of MMP-2 peak in simvastatin densitograms versus control densitogram, which served as 100% of MMP-2 activity. Inhibition of MMP-2 activity in gels with simvastatin was expressed as a percent of MMP-2 activity in control gels. The experiment was repeated three times (each time 10 samples in three gels) and the mean value was taken for comparisons. Standards of MMPs activity were used to identify gelatinases—standard from human capillary blood, which contains all forms of gelatinases [26] and supernatant from human megakaryocytes culture, which contains MMP-2 of 72 kDa and 68 kDa molecular weight.

### 2.6. Assessment of MMP-2, RhoA, Phosphorylated Myosin Light Chain Subunit 1 (Phospho-MYL9), and Troponin I Content by Western Blot

Samples containing 20 or 50 µg (for MMP-2, RhoA, and troponin I or phospho-MYL9, respectively) of protein obtained from heart extracts were mixed with 4x Laemmli Sample Buffer (BioRad, Hercules, CA, USA) with the addition of β-mercaptoethanol (1:10; *v*:*v*) and samples for RhoA were additionally denatured by 2 min incubation at 100 °C. All samples were applied to 10% SDS-PAGE gels prepared on the basis of TGX Stain-Free FastCast Acrylamide Kit 10% (BioRad, Hercules, CA, USA). After electrophoresis was completed (120 V, 20 °C), separated protein fractions were electroblotted by wet transfer onto a nitrocellulose membrane (50 V, 30 min). A primary monoclonal mouse antibody against total MMP-2 (ab86607, Abcam, Cambridge, Great Britain), a mouse monoclonal antibody against amino acids 120–150 of RhoA of human origin (sc-418, Santa Cruz Biotechnology, Inc., Dallas, TX, USA), a primary monoclonal mouse antibody against cardiac troponin I (MA1-20121, Thermo Fisher Scientifi, Waltham, MA USA), as well as secondary goat-anti-mouse conjugated with horseradish peroxidase (HRP) (BioRad, Hercules, CA, USA) were used at a dilution of 1:1000. A rabbit polyclonal IgG antibody against myosin light chain 2 (PA5-17727, Invitrogen, Rockford, IL, USA) was used at a dilution of 1:1000 and secondary goat-anti-rabbit conjugated with HRP (BioRad, Hercules, CA, USA) was used at a dilution of 1:2000. Blots were developed using chemiluminescence assay (ClarityTM Western ECL Substrate, BioRad, Hercules, CA, USA). Membranes were scanned using ChemiDocTM XRS+ System with Image LabTM Software v.5.2 for data analysis. Rat cardiac troponin I was used for standard curve preparation (Advanced ImmunoChemical Inc., Long Beach, CA, USA). Relative quantity of the other proteins of interest was expressed in arbitrary units (AU) and calculated using one of the samples serving as a standard with a given value of 1 AU in each gel. Each line contained exactly the same amount of protein which served as a normalization for comparisons. Protein molecular weight was assessed by comparison to Precision Plus Protein™ All Blue Standards (BioRad, Hercules, CA, USA).

### 2.7. Statistical Analysis

GraphPad Prism 7.0 software was used for the statistical analysis (La Jolla, CA, USA). All data are expressed as mean ± SEM. The analysis was performed with Kruskal–Wallis one-way analysis of variance followed by the post-hoc analysis (Dunn test with Bonferroni correction). In vitro MMP-2 inhibition experiment values were compared by RM one-way ANOVA with Tukey’s post-hoc analysis after excluding one set of data (outliers). *p*-values < 0.05 were considered statistically significant.

## 3. Results

### 3.1. Simvastatin Prevented from Cardiac Mechanical Dysfunction in Rat Hearts Subjected to Ischemia-Reperfusion

To determine if simvastatin has a protective effect on myocardial function, isolated heart perfusion was performed. The aerobically perfused hearts showed a stable mechanical function expressed as RPP throughout the perfusion period. In the I/R group, the mechanical function after 30 min of reperfusion was significantly impaired compared to the control group (7.6 ± 2.1 vs. 16.1 ± 2.5, *n* = 6–7, *p* < 0.05). In the group treated with simvastatin, the RPP at 30 min of reperfusion did not differ from the aerobically perfused hearts, which indicates the cardioprotective effect of simvastatin (Figure 1).

### 3.2. Simvastatin Inhibited MMP-2 Activity in Heart Tissue in Ischemia-Reperfusion Injury Model

We employed gelatin zymography to determine the activity of MMP-2 in heart tissue, since MMP-2 belongs to the proteases using gelatin as a substrate. As MMP-2 is synthesized as a pro-form further converted to its active form by proteolytic cleavage of its pro-domains, MMP-2 was split into two bands: 72 and 62 kDa. In the hearts treated with simvastatin exposed to I/Ri, 72 kDa MMP-2 activity was significantly reduced compared to the I/Ri group (160.7 ± 32.07 vs. 82.48 ± 36.56, *n* = 5–6, *p* < 0.05) (Figure 2a,b). This suggests that the drug has an inhibitory effect on MMPs activation under ischemia and reperfusion conditions.

### 3.3. Simvastatin Did Not Affect MMP-2 Content in Heart Tissue

To determine whether this inhibitory effect of simvastatin could be related with an increase of MMP-2 protein content in heart tissue, MMP-2 content was assessed by Western blot. There were no significant differences in the content of 72 kDa MMP-2 in heart tissue (Figure 3a,b). Thus, the reduction in MMP-2 activity is not due to a reduction in MMP-2 content in heart tissue.

### 3.4. Simvastatin Did Not Change MMP-2 mRNA Expression in Hearts Subjected to Ischemia-Reperfusion

We then investigated whether simvastatin was affecting MMP-2 at the mRNA level. Real time PCR revealed no significant changes in *MMP-2* mRNA expression between groups (*n* = 5–6) (Figure 4). Therefore, our data indicate that the inhibitory effect of simvastatin on MMP-2 does not affect mRNA but occurs at the posttranscriptional level.

### 3.5. Simvastatin Decreased MMP-2 Activity In Vitro

We next determined if simvastatin has a direct inhibitory effect on MMP-2. Simvastatin inhibited the activity of MMP-2 when run out on gel zymograms incubated with 3 and 30 µM simvastatin (Figure 5a,b). This suggests that the drug has a direct inhibitory effect on the activity of MMP-2.

### 3.6. Simvastatin Decreased RhoA Content in Hearts Subjected to Ischemia-Reperfusion

To determine whether simvastatin had an impact on the RhoA/ROCK pathway, RhoA content in heart tissue was assessed by Western blot. In simvastatin-treated hearts subjected to ischemia-reperfusion RhoA content was significantly lower in comparison to aerobically perfused hearts, while it did not differ between groups not treated with simvastatin (0.473 ± 0.064 vs. 0.291 ± 0.045, *n* = 5–6, *p* < 0.05) (Figure 6a,b). This indicates that simvastatin has an inhibitory effect on activity on RhoA.

### 3.7. Simvastatin Decreased Phosphorylated Myosin Light Chain Subunit 1 Content in Hearts Subjected to Ischemia-Reperfusion

To further support these observations, we assessed a downstream substrate of the RhoA/ROCK pathway—phosphorylated myosin light chain subunit 1 content in heart tissue. It was significantly lower in simvastatin-treated hearts subjected to is-chemia-reperfusion in comparison to aerobically perfused hearts while it did not differ between groups not treated with simvastatin (0.3960 ± 0.1569 vs. 0.9636 ± 0.2834 and 0.7542 ± 0.3101; *n* = 5–6, *p* < 0.05) (Figure 7a,b). The result indicates that the RhoA/ROCK pathway was inhibited after a reduction in RhoA content in heart tissue.

### 3.8. Simvastatin Has No Effect on Troponin Level in Hearts Subjected to Ischemia-Reperfusion

Finally, we investigated whether simvastatin was reducing troponin I degradation in heart tissue. Western blot analysis showed no significant differences in troponin I content in heart tissue (Figure 8a,b). Considering that troponin I is a part of thin filaments necessary for normal cardiac contraction and in keeping with our observation with simvastatin cardioprotective effect, this result suggests that the improvement in mechanical heart function by simvastatin is not due to its effect on troponin.

## 4. Discussion

The RhoA/ROCK signaling pathway activation is associated with the development of various cardiovascular pathological processes including cardiac hypertrophy, apoptosis as well as systemic and pulmonary hypertension [18].

The Rho kinase which is closely associated with cell apoptosis is ROCK1, which preferentially localizes to actomyosin filament bundles and in its active form phosphorylates MLC. This leads to apoptotic membrane blebbing [19,20]. The pro-apoptotic properties of ROCK1 were shown in the heart muscle [21,22]. Additionally, it was demonstrated in clinical studies that increased ROCK activity serves as a predictor of worsened outcomes in patients with myocardial infarction [23]. Various animal studies demonstrated that inhibition of Rho kinases by its inhibitors such as fasudil and statins have been shown to reduce apoptosis of myocardial cells and improve the impaired cardiac mechanical function in ischemia-reperfusion model or induced myocardial infarction in rats [24,25].

Statins such as simvastatin, lovastatin and atorvastatin were shown to downregulate phosphorylation of MYPT1 and MLC which are downstream substrates of RhoA/ROCK signaling pathway and to inhibit activation of MMP-2 and MMP-9 in astrocytes of the human optic nerve head [14]. It was also demonstrated that simvastatin reduced MMP-9 secretion from human saphenous vein smoot muscle cells by inhibiting the RhoA/ROCK pathway and decreasing MMP-9 mRNA levels [15]. However, there were no data on the effect of statins on MMP activation via RhoA/ROCK pathway in acute IR injury of the heart.

In our study we showed for the first time that simvastatin treatment caused improvement of contractile function of the heart subjected to ischemia-reperfusion, which was accompanied by reduction of MMP-2 activity in heart tissue along with inhibition of RhoA pathway expressed as a reduction of the content of both RhoA and its downstream product—MYL9-phospho in simvastatin-treated hearts.

These results are partially in line with findings of Kim et al. and Turner et al. [14,15]. Kim et al. showed that statins suppress MMP-2 and MMP-9 expression and activation through RhoA/ROCK pathway inhibition in astrocytes of the human optic nerve head [14]. Our current results are also partially consistent with previous data by Turner et al. showing that the simvastatin may reduce MMPs secretion by inhibiting the RhoA/ROCK pathway and decreasing MMP RNA levels [15]. They assessed the effect of statins, including simvastatin, on transforming growth factor-β2 (TGF-β2) mediated phosphorylation of MYPT1 and on MMP-2 and MMP-9 activities in cell culture. Cells were incubated with either TGF-β2 alone or with statins, followed by TGF-β2. The authors showed that statins significantly suppressed MYPT1 phosphorylation as well as enhanced by TGF-β2 MMP-2 and MMP-9 activation. As the authors previously reported that statins also suppress the TGF-β2-mediated expression of extracellular matrix molecules in human astrocytes of the human optic nerve head, they suggest that statins modulate TGF-β2 action by suppressing RhoA/ROCK signaling pathways and they speculate that the ability of statins to modulate TGF-β2 has an impact on MMPs activities. Turner et al. showed that the simvastatin significantly reduced 12-O-tetradecanoylphorbol 13-acetate and platelet-derived growth factor-BB/Interleukine-1-induced MMP-9 activation in organ-cultured human saphenous vein by inhibiting the RhoA/ROCK pathway and decreasing *MMP* RNA levels [15]. They suggest that RhoA/ROCK inhibition leads to a reduction in *MMP-9* mRNA levels at the transcriptional level, possibly as a result of reduced transcription factor binding.

Opposite to Turner et al. findings, in our study we did not observe differences in *MMP-2* mRNA expression between groups, indicating that we can exclude changes of MMP-2 at the transcriptional level. This may be related with relatively short time of exposure for simvastatin during ischemia-reperfusion, not enough to cause changes in mRNA expression. Together with Western blot analysis of MMP-2 content in heart tissue, which revealed that there were no changes between groups, the data indicate that the decrease in MMP-2 activity by simvastatin was due to its downregulation on a posttranscriptional level. One of the possible mechanisms may be that postulated by Kim et al.: the inhibitory effect of simvastatin on the activity of TGF-β2, however, in the I/R injury, TGF-β2 is not the main activating factor of metalloproteinases and therefore its inhibition should only partially affect the activity of MMP-2. Another possible mechanism of the inhibitory effect of simvastatin on the post-transcriptional activation of MMP-2 is its direct inhibitory effect, which was demonstrated in the in vitro experiment. Based on the results of our in vitro experiment which showed direct effect of simvastatin on MMP-2 activity we may conclude that the inhibition of MMP activity along with inhibition of RhoA pathway protein content in the settings of acute ischemia-reperfusion may be a coincidence or only partly due to the relationship between these pathways. This hypothesis is partially confirmed by our demonstration that degradation of troponin I in the settings of I/R was not inhibited. Since troponin I is one of the final substrates for MMP-2, the result indicates that the reduction in MMP-2 activity by simvastatin was insufficient to inhibit troponin degradation. As troponin I is part of the thin filaments necessary for normal cardiac contraction; this result suggests that the improvement of the mechanical heart function by simvastatin is not due to its effect on troponin via downregulation of MMP-2. It may suggest that the main mechanism responsible for the protective effect of simvastatin on the ischemic and reperfused heart muscle does not result from the inhibition of MMP-2 activity, but other mechanisms related with inhibition of RhoA pathway activation.

Unfortunately, based on our data, we cannot directly say what is the exact mechanism responsible for the inhibition of MMP-2 and for the protection of the heart contractile function by simvastatin, which is the main limitation of our study and requires further detailed research. Demonstration that simvastatin has a direct effect on MMP2 activity, RhoA/ROCK or MYPT1/MLC overexpression in a cell culture model and testing of MMP2 at all levels is required. It is possible that simvastatin inhibits protease MT1-MMP, therefore having an additional effect beyond direct action on MMP2 activity revealed in our in vitro experiment. Thus, MT1-MMP should be tested in both the hearts and cell culture model systems. Additionally, TGF-β2 activity should be tested. Finally, the relationship between inhibition of the RhoA/ROCK pathway and inhibition of MMP-2 activation by simvastatin in the presence of another RhoA inhibitor not affecting MMP-2 should be examined.

In light of our previous findings and the studies to date, we may conclude that inhibition of the of MMP-2 activation by simvastatin in ischemia-reperfusion model, although it is accompanied by a decrease of Rho-A and MYL9-phospho content, does not result from an inhibition of *MMP-2* m-RNA synthesis resulting from an inhibition of the RhoA/ROCK pathway and is due, at least in part, to the direct effect of the drug. Moreover, the protective action of simvastatin on heart contractile function in an acute ischemia-reperfusion model seems not to be related with decreased MMP-2 activation but other mechanisms related with inhibition of the RhoA/ROCK pathway. Further research is needed to confirm this phenomenon, but our results, although preliminary, are of great importance as they constitute the first attempt to assess the relationship between two important pathogenetic pathways resulting from MMP-2 and RhoA activation, leading to ischemia and reperfusion damage to the heart muscle. This opens up new perspectives in understanding the physiology of I/R damage to the heart muscle and exploring new therapeutic options.

## Figures and Tables

**Figure 1 biomolecules-12-01291-f001:**
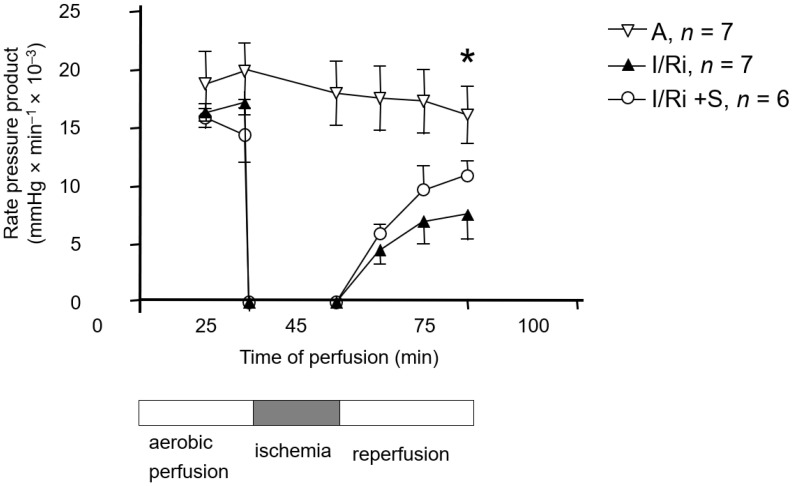
Effect of simvastatin on cardiac mechanical function after 20 min of ischemia. Result presented as the rate-pressure product (heart rate × left ventricular developed pressure) during reperfusion. Cardiac mechanical function at 30 min of reperfusion in I/R group was significantly impaired in comparison to control group but did not differ from the one of aerobically perfused hearts in simvastatin-treated group. A—aerobic perfusion, IRi—ischemia-reperfusion injury, IRi + S—hearts from IR injury model treated with simvastatin. * *p* < 0.05, *n* = 6–7, ANOVA.

**Figure 2 biomolecules-12-01291-f002:**
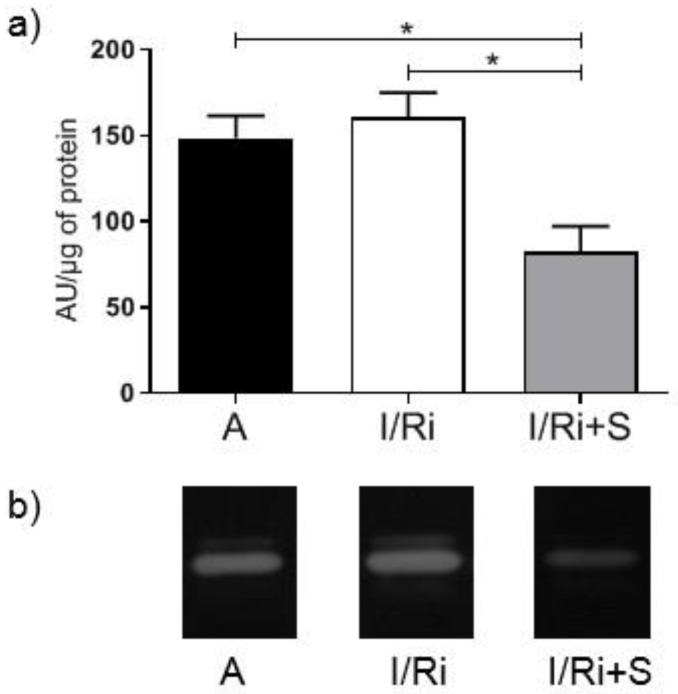
Matrix metalloproteinase 2 (MMP-2) activity in hearts subjected to ischemia-reperfusion (**a**) The MMP-2 activity in heart tissue samples subjected to I/R. Gelatin zymography showed significant differences in MMP-2 activity in hearts of group treated with simvastatin compared to I/Ri group without simvastatin. MMP-2 activity expressed in arbitrary units per µg of protein. * *p* < 0.05, *n* = 5–6, ANOVA. (**b**) Representative zymogram showing MMP-2 gelatinolytic activity in heart tissue. A—aerobic perfusion, I/Ri—ischemia-reperfusion injury, I/Ri + S—hearts from IR injury model treated with simvastatin.

**Figure 3 biomolecules-12-01291-f003:**
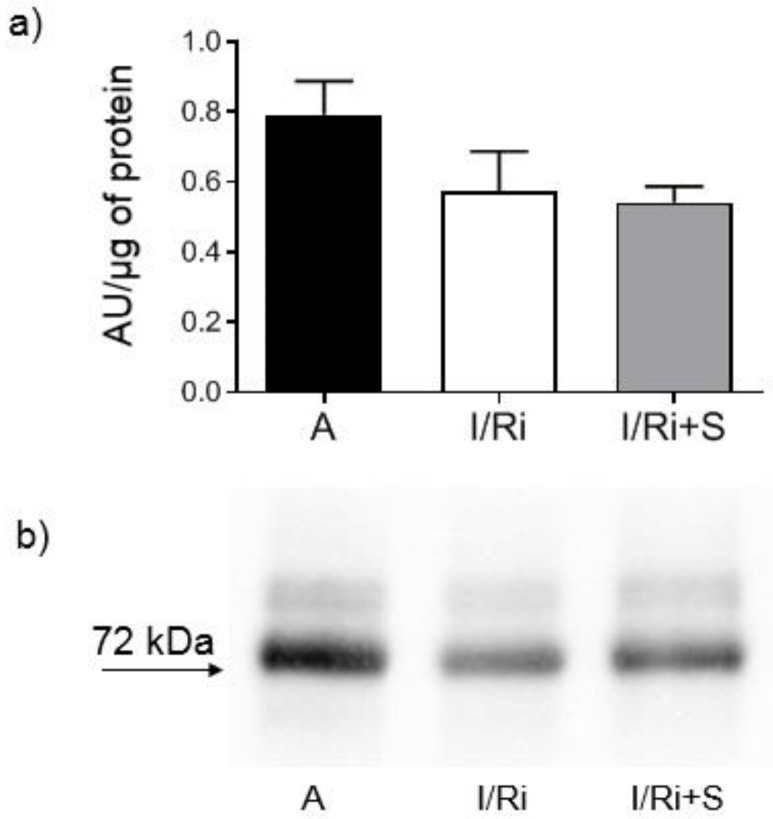
Effect of simvastatin on MMP-2 content in hearts subjected to ischemia-reperfusion (**a**) Analysis of MMP-2 content in heart tissue homogenates determined by Western blot showed no differences. MMP-2 expressed in arbitrary units per µg of protein. *n* = 5–6, ANOVA (**b**) Representative blot showing MMP-2 level in heart tissue homogenates. A—aerobic perfusion, I/Ri—ischemia-reperfusion injury, I/Ri + S—hearts from IR injury model treated with simvastatin. *n* = 5–6, ANOVA.

**Figure 4 biomolecules-12-01291-f004:**
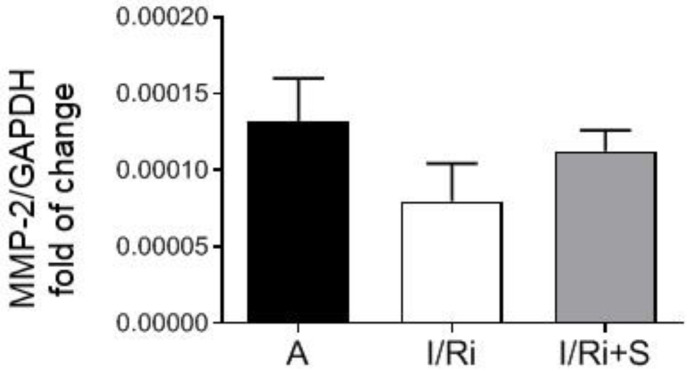
*MMP-2* mRNA expression normalized to glyceraldehyde-3-phosphate dehydrogenase (*GAPDH*) reference gene in hearts subjected to ischemia-reperfusion. Real time PCR showed no significant changes in *MMP-2* mRNA expression in heart tissue homogenates between groups. *n* = 5–6, ANOVA. A—aerobic perfusion, I/Ri—ischemia-reperfusion injury, I/Ri + S—hearts from IR injury model treated with simvastatin.

**Figure 5 biomolecules-12-01291-f005:**
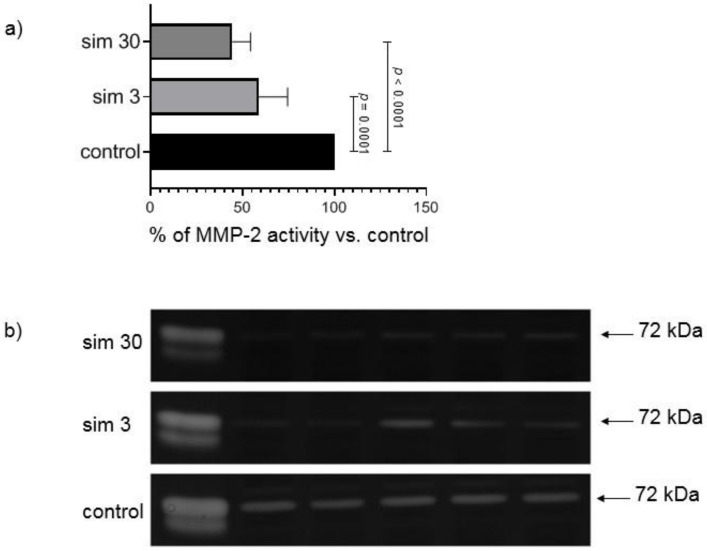
MMP-2 activity in in vitro experiment. (**a**) The 72 kDa MMP-2 specific activity in heart samples from groups not pre-treated with simvastatin placed in zymography incubation buffer containing DMSO (control) or the addition of simvastatin stock solutions to achieve final concentration 3 μmol/L (sim 3) or 30 μmol/L (sim 30). Simvastatin inhibited the activity of MMP-2 when run out on gel zymograms incubated with 3 and 30 µM simvastatin. *n* = 9; ANOVA with Tukey’s post-hoc test. (**b**) Representative zymograms of gelatinolytic MMP-2 activity in the same heart samples placed either in control buffer or buffer with addition of simvastatin stock solutions (sim 3 and sim 30).

**Figure 6 biomolecules-12-01291-f006:**
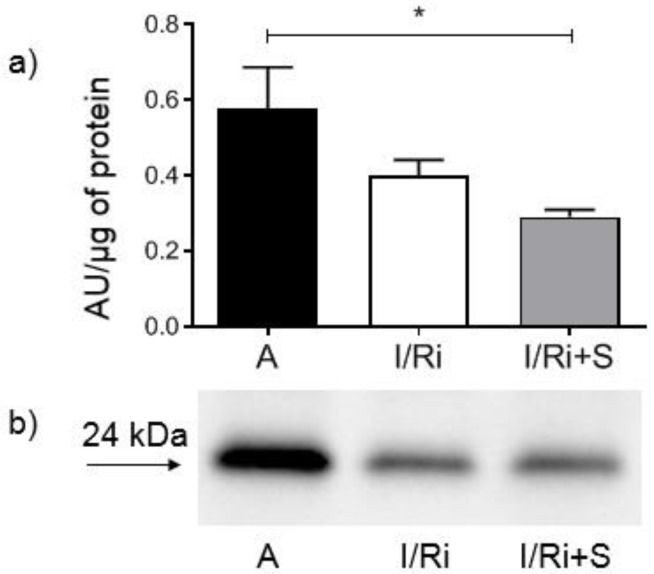
Effect of simvastatin on RhoA content in hearts subjected to ischemia-reperfusion (**a**) Analysis of RhoA content in heart homogenates determined by Western blot. RhoA content in heart tissue in simvastatin-treated hearts subjected to ischemia-reperfusion was significantly lower in comparison to aerobically perfused hearts, while it did not differ between groups not treated with simvastatin. * *p* < 0.05, *n* = 5–6, ANOVA. (**b**) Representative Western blot showing RhoA level in heart homogenates. A—aerobic perfusion, I/Ri—ischemia-reperfusion injury, I/Ri + S—hearts from IR injury model treated with simvastatin.

**Figure 7 biomolecules-12-01291-f007:**
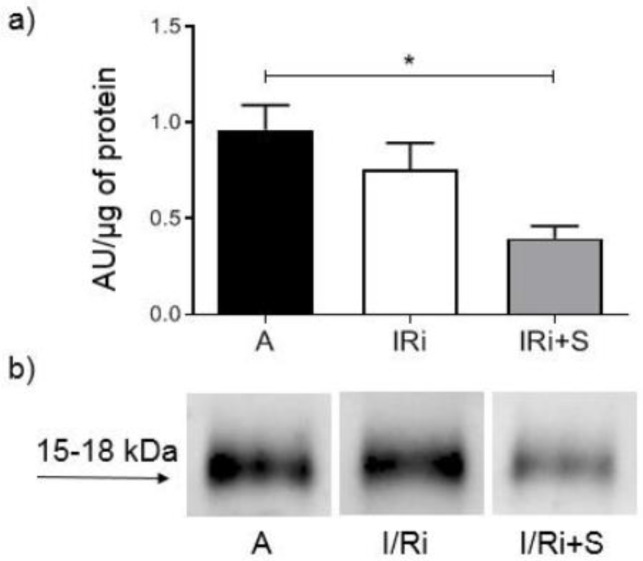
Effect of simvastatin on phosphorylated myosin light chain subunit 1 content in hearts subjected to ischemia-reperfusion. (**a**) Analysis of phosphorylated myosin light chain subunit 1 content in heart homogenates determined by Western blot. Phosphorylated myosin light chain subunit 1 content in heart tissue in simvastatin-treated hearts subjected to ischemia-reperfusion was significantly lower in comparison to aerobically perfused hearts, while it did not differ between groups not treated with simvastatin. * *p* < 0.05, *n* = 5–6, ANOVA. (**b**) Representative blot showing phosphorylated myosin light chain subunit 1 level in heart homogenates. A—aerobic perfusion, I/Ri—ischemia-reperfusion injury, I/Ri + S—hearts from IR injury model treated with simvastatin.

**Figure 8 biomolecules-12-01291-f008:**
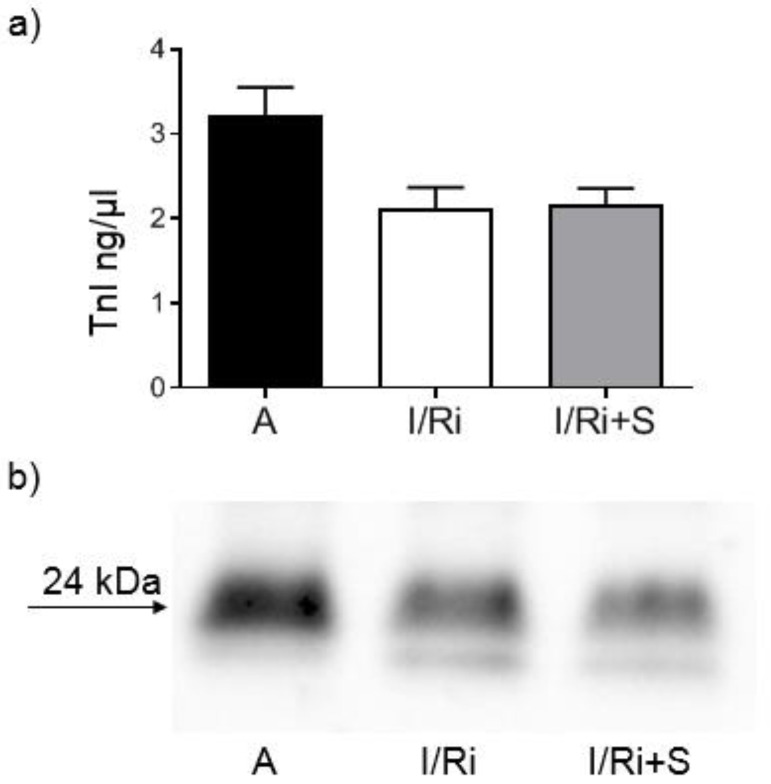
Effect of simvastatin on Troponin I content in hearts subjected to ischemia-reperfusion. (**a**) Analysis of Troponin I content in heart homogenates determined by Western blot. There were no significant differences in troponin I content in heart tissue. *n* = 5–6, ANOVA. (**b**) Representative blot showing the Troponin I level in heart homogenates determined by Western blot. A—aerobic perfusion, I/Ri—ischemia-reperfusion injury, I/Ri + S—hearts from IR injury model treated with simvastatin.

## Data Availability

Data available upon request.

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
