# Peer review of "The Protective Effect of Simvastatin on the Systolic Function of the Heart in the Model of Acute Ischemia and Reperfusion Is Due to Inhibition of the RhoA Pathway and Independent of Reduction of MMP-2 Activity"

_biomolecules, 2022, doi:10.3390/biom12091291_

Round 1

Reviewer 1 Report

The authors aimed to establish whether another RhoA/ROCK signaling pathway inhibitor simvastatin lowers MMP-2 activity in ischemia-reperfusion injury rat model by inhibiting the RhoA/ROCK pathway and reducing MMP-2 mRNA levels.

-       Methods are well structured and Statistical analysis was rigorously performed.

-       The study showed that simvastatin treatment caused improvement of contractile function of the heart subjected to I/Ri which was accompanied by reduction of MMP-2 activity in heart tissue along with inhibition of RhoA pathway expressed as a reduction of the content of both RhoA and its downstream product – phospho-MYL9 in simvastatin treated hearts.

-       Figures are clear and very informative.

-       The Discussion section is the part that requires some mandatory amendments from the authors. In fact, in the current format, the discussion is too short and superficially written it lacks a critical point of view also in the context of other papers published in the current literature.

-       Moreover, the paper needs to be revised also for English style.

Thanks for the opportunity to review and best wishes with your paper.

Author Response

Reviewer #1 comments:

The authors aimed to establish whether another RhoA/ROCK signaling pathway inhibitor simvastatin lowers MMP-2 activity in ischemia-reperfusion injury rat model by inhibiting the RhoA/ROCK pathway and reducing MMP-2 mRNA levels.

-       Methods are well structured and Statistical analysis was rigorously performed.

-       The study showed that simvastatin treatment caused improvement of contractile function of the heart subjected to I/Ri which was accompanied by reduction of MMP-2 activity in heart tissue along with inhibition of RhoA pathway expressed as a reduction of the content of both RhoA and its downstream product – phospho-MYL9 in simvastatin treated hearts.

-       Figures are clear and very informative.

Response:

We thank Reviewer #1 for appreciative comments of our work. Please find below our answers to the specific comments. 

-       The Discussion section is the part that requires some mandatory amendments from the authors. In fact, in the current format, the discussion is too short and superficially written it lacks a critical point of view also in the context of other papers published in the current literature.

Response:

We greatly appreciate Reviewer #1 for comments. Suggested changes have been done and they have significantly improved the discussion section. In response to Reviewer #1 comment the text on pages 12–13, lines 442–443, 447–462, 466–509, 517–523 was changed as follows:

„These results are partially in line with findings of Kim at al. and Turner at al [14,15]. Kim at al. showed that statins suppress MMP-2 and MMP-9 expression and activation through RhoA/ROCK pathway inhibition in astrocytes of the human optic nerve head [14]. Our current results are also partially consistent with previous data by Turner at all. showing that the simvastatin may reduce MMPs secretion by inhibiting the RhoA/ROCK pathway and decreasing MMP RNA levels [15]. They assessed the effect of statins, including simvastatin, on transforming growth factor-β2 (TGF-β2) mediated phosphorylation of MYPT1 and on MMP-2 and MMP-9 activities in cell culture. Cells were incubated with either TGF-β2 alone or with statins, followed by TGF-β2. The authors showed that statins significantly suppressed MYPT1 phosphorylation as well as enhanced by TGF-β2 MMP-2 and MMP-9 activation. As the authors previously reported that statins also suppress the TGF-β2-mediated expression of extracellular matrix molecules in human astrocytes of the human optic nerve head they suggest that statins modulate TGF-β2 action by suppressing RhoA/ROCK signaling pathways and they speculate that the ability of statins to modulate TGF-β2 has an impact on MMPs activities. Turner at al. showed that the simvastatin significantly reduced 12-O-tetradecanoylphorbol 13-acetate and platelet-derived growth factor-BB/Interleukine-1-induced MMP-9 activation in organ-cultured human saphenous vein by inhibiting the RhoA/ROCK pathway and decreasing MMP RNA levels [15]. They suggest that RhoA/ROCK inhibition leads to a reduction in MMP-9 mRNA levels at the transcriptional level, possibly as a result of reduced transcription factor binding.

Opposite to Turner at al. findings, in our study we did not observe differences in MMP-2 mRNA expression between groups, indicating that we can exclude changes of MMP-2 at the transcriptional level. This may be related with relatively short time of exposure for simvastatin during ischemia-reperfusion, not enough to cause changes in mRNA expression. Together with Western blot analysis of MMP-2 content in heart tissue which revealed that there were no changes between groups the data indicate that the decrease in MMP-2 activity by simvastatin was due to its downregulation on posttranscriptional level. One of the possible mechanisms may be the postulated by Kim et al. inhibitory effect of simvastatin on the activity of TGF-β2, however, in the I/R injury, TGF-β2 is not the main activating factor of metalloproteinases and therefore its inhibition should only partially affect the activity of MMP-2. Another possible mechanism of the inhibitory effect of simvastatin on the post-transcriptional activation of MMP-2 is its direct inhibitory effect, which was demonstrated in the in-vitro experiment. Based on the results of our in vitro experiment which showed direct effect of simvastatin on MMP-2 activity we may conclude that the inhibition of MMP activity along with inhibition of RhoA pathway activity in the settings of acute ischemia-reperfusion may be a coincidence or only partly due to the relationship between these pathways. This hypothesis is partially confirmed by our demonstration that degradation of troponin I in the settings of I/R was not inhibited. Since troponin I is one of the final substrates for MMP-2, the result indicates that the reduction in MMP-2 activity by simvastatin was insufficient to inhibit troponin degradation. As troponin I is part of the thin filaments necessary for normal cardiac contraction, this result suggests that the improvement of the mechanical heart function by simvastatin is not due to its effect on troponin via downregulation of MMP-2. It may suggest that the main mechanism responsible for the protective effect of simvastatin on the ischemic and reperfused heart muscle does not result from the inhibition of MMP-2 activity, but other mechanisms related with inhibition of RhoA pathway activation.

Unfortunately, based on our data, we cannot directly say what is the exact mechanism responsible for the inhibition of MMP-2 and for the protection of the heart contractile function by simvastatin which is the main limitation of our study and requires further  detailed research. Demonstration that simvastatin has a direct effect on MMP2 activity, RhoA/ROCK or MYPT1/MLC overexpression in a cell culture model and testing of MMP2 at all levels is required. It is possible that simvastatin inhibits protease MT1-MMP therefore having an additional effect beyond direct action on MMP2 activity revealed in our in vitro experiment. Thus, MT1-MMP should be tested in both the hearts and cell culture model systems. Also TGF-β2 activity should be tested. Finally, the relationship between inhibition of the RhoA-ROCK pathway and inhibition of MMP-2 activation by simvastatin in the presence of another RhoA inhibitor not affecting MMP-2 should be examined.

In the light of our previous findings and the studies to date, we may conclude that inhibition of the of MMP-2 activation by simvastatin in ischemia-reperfusion model, although it is accompanied by decrease of Rho-A and MYL9-phospho content, does not result from an inhibition of MMP-2 m-RNA synthesis resulting from an inhibition of RhoA/ROCK pathway and is due, at least in part, to the direct effect of the drug. Moreover, the protective action of simvastatin on heart contractile function in acute ischemia-reperfusion model seems not to be related with decreased MMP-2 activation but other mechanisms related with inhibition RhoA/ROCK pathway. Further research is needed to confirm this phenomenon, but our results, although preliminary, are of great importance as they constitute the first attempt to assess the relationship between two important pathogenetic pathways resulting from MMP-2 and RhoA activation leading to ischemia and reperfusion damage to the heart muscle. This opens up new perspectives in understanding the physiology of I/R damage to the heart muscle and exploring new therapeutic options.”

-       Moreover, the paper needs to be revised also for English style.

Response:

In response to Reviewer #1 comment the text underwent English revision. All changes were introduced in “Track changes” mode.

Thanks for the opportunity to review and best wishes with your paper.

Response:

We greatly appreciate this comment. Changes suggested have been done and we do hope that the manuscript is now acceptable for publication in Biomolecules, Special Issue "Matrix Metalloproteinases in Health and Disease 3.0".

Reviewer 2 Report

This study shows that simvastatin increases heart contractile function while also decreasing MMP2 activity, although it is stated that the beneficially outcome of simvastatin does not involve MMP2. They have previously shown that simvastatin inhibits activation of MMP2 through downregulation of MYPT1 and MLC phosphorylation. The present study aims to see if the inhibition is also direct.

The manuscript lacks in description of the data with usually only 1 sentence describing the figure in the section. Therefore, the data is not fully described and put into relevant context for the preceding and following data. This study seemingly has two sets of data (heart tissue and in vitro model) that were used to perform a few experiments consisting of a few western blots, two MMP2 zymograms and one qRT-PCR experiment. More data to discern the key findings is suggested.   

The effect of simvastatin on MMP2 activity was already shown in astrocytes and is somewhat expected in acute IR heart injury. The data herein only add mildly to the literature with the major finding showing that MMP2 activity is downregulated, but the authors conclude that this seemingly has no effect on the beneficial outcomes of simvastatin. Please see the concerns listed below.

Major:

MMP2 requires proteolytic processing for activation (PMID: 18974156). In the cell culture model, MMP2 should be activated, in the presence of simvastatin, by a protease (MT1-MMP/MMP14) and follow a readout of MMP2 activity. It is possible that simvastatin inhibits MT1-MMP and therefore does not have a direct effect on MMP2 activity. This would explain the puzzling no change in MMP2 mRNA levels presented in the study. Therefore, it is suggested to test MT1-MMP in both the hearts and cell culture model systems.

Additionally, to show that simvastatin causes a direct effect on MMP2 activity, the authors could overexpress RhoA/ROCK, or MYPT1/MLC in the cell culture model and interrogate MMP2 at all levels.

Figure 7: The western blot is somewhat weak as compared to published data with the antibody. It is suggested to redo the western blot and obtain data that will visually show the apparent difference.

It is suggested to obtain results of MMP2 mRNA, RhoA protein, etc. from the in vitro assay. At present, only MMP2 activity is analyzed. This would assist in fully backing up the in vivo data obtained.

Line 403-406: This sentence needs to be explained in more detail. If one cannot distinguish between RhoA/ROCK pathway inhibition and decreased MMP2 activation, how can it be concluded that the protective effects of simvastatin are not related to reduced MMP2 activity, but rather to inhibition of RhoA/ROCK?

Minor:

Line 26: Should activation read “inactivation”?

Figure 3: A housekeeping gene is needed and the percentage of MMP2 over loading control should be quantified, not just the MMP2 bands.

Line 244 and others: MMP2 should be italicized (all mRNA should be italicized).

Line 290: A period missing at the end of the figure legend.

Line 377 and 380: “at all.” should read “et al.”

Line 389: “RhoA pathway activity” should read RhoA pathway protein content”.

Line 400: Remove the second “of”.

Author Response

Reviewer # 2

This study shows that simvastatin increases heart contractile function while also decreasing MMP2 activity, although it is stated that the beneficially outcome of simvastatin does not involve MMP2. They have previously shown that simvastatin inhibits activation of MMP2 through downregulation of MYPT1 and MLC phosphorylation. The present study aims to see if the inhibition is also direct.

The manuscript lacks in description of the data with usually only 1 sentence describing the figure in the section. Therefore, the data is not fully described and put into relevant context for the preceding and following data. This study seemingly has two sets of data (heart tissue and in vitro model) that were used to perform a few experiments consisting of a few western blots, two MMP2 zymograms and one qRT-PCR experiment. More data to discern the key findings is suggested.  

Response:

We thank the Reviewer # 2 for very careful review and all valuable comments, which helped improve the manuscript.

As suggested by reviewer # 2, the results section has been changed as follows (page 6; lines 211–220, 230–238; page 7; lines 256–261; page 8; lines 279 –282, 292–295; page 9; lines 325–333; page 10; lines 355–364; page 11; lines 385–390):

„To determine if simvastatin has a protective effect on myocardial function the isolated heart perfusion was performed. The aerobically perfused hearts showed a stable mechanical function expressed as RPP throughout the perfusion period. In the I/R group, the mechanical function after 30 minutes of reperfusion was significantly impaired compared to the control group (7.6 ± 2.1 vs 16.1 ± 2.5, N = 5, p <0.05). In the group treated with simvastatin, the RPP at 30 minutes of reperfusion did not differ from the aerobically perfused hearts, which indicates the cardioprotective effect of simvastatin (Figure 1)”.

“We employed gelatin zymography to determine the activity of MMP-2 in heart tissue, since MMP-2 belongs to the proteases using gelatin as a substrate. As MMP-2 is synthesized as a pro-form further converted to its active form by proteolytic cleavage of its pro-domains, MMP-2 was split into two bands: 72 and 62 kDa. In the hearts treated with simvastatin exposed to I/Ri, 72 kDa MMP-2 activity was significantly reduced compared to the I/Ri group (160.7±32.07 vs 82.48±36.56, N=5-6, p<0.05) (Figure 2 a, b). This suggests that the drug has an inhibitory effect on MMPs activation under ischemia and reperfusion conditions”.

“To determine whether this inhibitory effect of simvastatin could be related with an increase of MMP-2 protein content in heart tissue, MMP-2 content was assessed by Western blot. There were no significant differences in the content of 72 kDa MMP-2 in heart tissue (Figure 3 a, b). Thus, the reduction in MMP-2 activity is not due to a reduction in MMP-2 content in heart tissue”.

“We then investigated whether simvastatin was affecting MMP-2 at the mRNA level. Real time PCR revealed no significant changes in MMP-2 mRNA expression between groups (N=5–6) (Figure 4). Therefore our data indicate that the inhibitory effect of simvastatin on MMP-2 does not affect mRNA but occurs at the posttranscriptional level”.

“We next determined if simvastatin has a direct inhibitory effect on MMP-2. Simvastatin inhibited the activity of MMP-2 when run out on gel zymograms incubated with 3 and 30 µM simvastatin (Figure 5 a, b). This suggests that the drug has a direct inhibitory effect on the activity of MMP-2”.

“To determine whether simvastatin had an impact on the RhoA-ROCK pathway, RhoA content in heart tissue was assessed by Western blot. In simvastatin treated hearts subjected to ischemia-reperfusion RhoA content was significantly lower in comparison to aerobically perfused hearts, while it did not differ between groups not treated with simvastatin  (0.473±0.064 vs 0.291±0.045, N=5–6, p<0.05) (Figure 6 a, b). This indicates that simvastatin has an inhibitory effect on activity on RhoA”.

“To  further  support  these  observations, we assessed a downstream substrate of the RhoA-ROCK pathway – phosphorylated myosin light chain subunit 1 content in heart tissue. It was significantly lower in simvastatin treated hearts subjected to is-chemia-reperfusion in comparison to aerobically perfused hearts while it did not differ between groups not treated with simvastatin (0.3960±0.1569 vs. 0.9636±0.2834 and 0.7542±0.3101) N=5–6, p<0.05) (Figure 7 a, b). The result indicates that the RhoA-ROCK pathway was inhibited after a reduction in RhoA content in heart tissue”.

„Finally, we investigated whether simvastatin was reducing troponin I degradation in heart tissue. Western blot analysis showed no significant differences in troponin I content in heart tissue (Figure 8 a, b). Considering that troponin I is a part of thin filaments necessary for normal cardiac contraction and in keeping with our observation with simvastatin cardioprotective effect, this result suggests that the improvement in mechanical heart function by simvastatin is not due to its effect on troponin.”

The effect of simvastatin on MMP2 activity was already shown in astrocytes and is somewhat expected in acute IR heart injury. The data herein only add mildly to the literature with the major finding showing that MMP2 activity is downregulated, but the authors conclude that this seemingly has no effect on the beneficial outcomes of simvastatin. Please see the concerns listed below.

Response:

We are very grateful for this valuable comment. This remark takes our attention to the fact that we have not emphasized enough what new data this work brings to the literature. Accordingly, the discussion has been modified as below (page 13, lines 517–523).

„Further research is needed to confirm this phenomenon, but our results, although preliminary, are of great importance as they constitute the first attempt to assess the relationship between two important pathogenetic pathways resulting from MMP-2 and RhoA activation leading to ischemia and reperfusion damage to the heart muscle. This opens up new perspectives in understanding the physiology of I/R damage to the heart muscle and exploring new therapeutic options.”

Please find below our answers to the specific comments. 

Major:

MMP2 requires proteolytic processing for activation (PMID: 18974156). In the cell culture model, MMP2 should be activated, in the presence of simvastatin, by a protease (MT1-MMP/MMP14) and follow a readout of MMP2 activity. It is possible that simvastatin inhibits MT1-MMP and therefore does not have a direct effect on MMP2 activity. This would explain the puzzling no change in MMP2 mRNA levels presented in the study. Therefore, it is suggested to test MT1-MMP in both the hearts and cell culture model systems.

Response:

We thank the  Reviewer #2 for an interesting suggestion. Unfortunately, because we did not performed cell culture, time given to us for revision of the manuscript is too short for performing new experiments and incorporating them to the manuscript (starting new cell culture and performing experiments will take few months). Moreover, the proteolysis is not the only one way to activate MMPs, as an example – gelatin zymography used in this study is an assay based on non-proteolytic activation of gelatinases in the presence of sodium dodecyl sulphate. Moreover, in this assay in the “in vitro” setting only the direct effect of simvastatin on MMPs can be observed as there is a lack of other factors influencing MMPs activity in the gel incubated in the buffer after electrophoresis. The reviewer’s comment was therefore acknowledged and included as a study limitation in the revised manuscript (page 13; lines 498–509):

„Unfortunately, based on our data, we cannot directly say what is the exact mechanism responsible for the inhibition of MMP-2 and for the protection of the heart contractile function by simvastatin which is the main limitation of our study and requires further detailed research. Demonstration that simvastatin has a direct effect on MMP2 activity, RhoA/ROCK or MYPT1/MLC overexpression in a cell culture model and testing of MMP2 at all levels is required. It is possible that simvastatin inhibits protease MT1-MMP therefore having an additional effect beyond direct action on MMP2 activity revealed in our in vitro experiment. Thus, MT1-MMP should be tested in both the hearts and cell culture model systems. Also TGF-β2 activity should be tested. Finally, the relationship between inhibition of the RhoA-ROCK pathway and inhibition of MMP-2 activation by simvastatin in the presence of another RhoA inhibitor not affecting MMP-2 should be examined”.

Additionally, to show that simvastatin causes a direct effect on MMP2 activity, the authors could overexpress RhoA/ROCK, or MYPT1/MLC in the cell culture model and interrogate MMP2 at all levels.

Response:

We thank Reviewer #1 for the suggestions. As we previously stated, it is a very important question and the main limitation of the study. Unfortunately, time given to us for revision of the manuscript is too short for performing new experiments and incorporating them to the manuscript. We have now acknowledged this as a study limitation and suggested it as a topic for further research in the Discussion section of the revised manuscript (page 13, lines 498–509).

„Unfortunately, based on our data, we cannot directly say what is the exact mechanism responsible for the inhibition of MMP-2 and for the protection of the heart contractile function by simvastatin which is the main limitation of our study and requires further  detailed research. Demonstration that simvastatin has a direct effect on MMP2 activity, RhoA/ROCK or MYPT1/MLC overexpression in a cell culture model and testing of MMP2 at all levels is required. It is possible that simvastatin inhibits protease MT1-MMP therefore having an additional effect beyond direct action on MMP2 activity revealed in our in vitro experiment. Thus, MT1-MMP should be tested in both the hearts and cell culture model systems. Also TGF-β2 activity should be tested. Finally, the relationship between inhibition of the RhoA-ROCK pathway and inhibition of MMP-2 activation by simvastatin in the presence of another RhoA inhibitor not affecting MMP-2 should be examined”.

Figure 7: The western blot is somewhat weak as compared to published data with the antibody. It is suggested to redo the western blot and obtain data that will visually show the apparent difference.

Response:

We thank Reviewer #1 for the suggestions. As suggested we performed western blotting for this protein in different conditions from freshly prepared samples with the use of phosphatases inhibitors. It improved the quality of the figure. We also recalculated the data although the results remain the same. The changes were introduced in “Methods” and “Results” sections (page 4, lines 118–120; page 5; line 181; page 10; lines 362–373)

It is suggested to obtain results of MMP2 mRNA, RhoA protein, etc. from the in vitro assay. At present, only MMP2 activity is analyzed. This would assist in fully backing up the in vivo data obtained.

Response:

We thank Reviewer #1 for the suggestion. It would be really informative to obtain such data. Unfortunately we didn’t find a way to perform the same type of experiment for other proteins as was performed specifically for MMPs activity. Our “in vitro” zymography experiment is based on the fact, that gelatinases possess gelatinolytic activity after SDS-PAGE electrophoresis while gelatin-containing gels are placed in the incubation buffer for 18 hours. The addition of simvastatin to the buffer can show if the direct inhibition of the enzyme is present. Other analysed proteins can not be made visible and analysed in similar method. We added the description of the in vitro assay to emphasize the background of the assay. The other in vitro experiments that could be more specific i.e. cell culture model, are unavailable for this study as mentioned above.

Changes were made on pages 4­–5, lines 157–159 as follows.

“This experiment was performed to assess the direct influence of simvastatin on gelatinolytic activity of MMPs separated during electrophoresis on polyacrylamide gelatin-containing gels by incubation of zymograms in the buffer containing the drug”.

Line 403-406: This sentence needs to be explained in more detail. If one cannot distinguish between RhoA/ROCK pathway inhibition and decreased MMP2 activation, how can it be concluded that the protective effects of simvastatin are not related to reduced MMP2 activity, but rather to inhibition of RhoA/ROCK?

Response:

Thank you for pointing it out. To make it more clear, the more specific explanation was included on page 11; lines 387–390.

„Considering that troponin I is a part of thin filaments necessary for normal cardiac contraction and in keeping with our observation with simvastatin cardioprotective effect, this result suggests that the improvement in mechanical heart function by simvastatin is not due to its effect on troponin”.

and in page 13; lines 481–488.

„Since troponin I is one of the final substrates for MMP-2, the result indicates that the reduction in MMP-2 activity by simvastatin was insufficient to inhibit troponin degradation. As troponin I is part of the thin filaments necessary for normal cardiac contraction, this result suggests that the improvement of the mechanical heart function by simvastatin is not due to its effect on troponin via downregulation of MMP-2. It may suggest that the main mechanism responsible for the protective effect of simvastatin on the ischemic and reperfused heart muscle does not result from the inhibition of MMP-2 activity, but other mechanisms related with inhibition of RhoA pathway activation”.

Minor:

Line 26: Should activation read “inactivation”?

Response:

The sentence was changed as below (page 2; line 27)

“MMP-2 inactivation is not due to inhibition of MMP-2 m-RNA synthesis caused by inhibition of RhoA/ROCK pathway and is due, at least in part, to the direct drug action.

Figure 3: A housekeeping gene is needed and the percentage of MMP2 over loading control should be quantified, not just the MMP2 bands.

Response:

We thank Reviewer #1 for this valuable suggestion. It would be additional control to enhance the reliability of our results. We didn’t use normalization protein in each blot as we used antibodies against housekeeping protein (tubulin) for validation of western blots on rat heart tissue in the setting of ischemia-reperfusion injury in previous experiments with simvastatin in our laboratory and we observed high correlation between the amount of loaded protein with the bands’ intensity and volume in densitogram. Thus we decided to use the total amount of protein per well as a sufficient way of normalization of results and we expressed them in arbitrary units. We added information on page 5, lines 198–199. Additional experiments with the use of normalization protein on each blot require more time and we were not able to perform them in such a short time.

Line 244 and others: MMP2 should be italicized (all mRNA should be italicized).

Response:

We thank Reviewer #1 for this comment. We made changes on page 2, lines 18, 22, and 28; page 3, lines 66, 86; page 4, lines 136–137, 139–141; page 8, lines 280, 290 (figure legend); page 12, lines 459, 460, 463; page 13, line 513.

Line 290: A period missing at the end of the figure legend.

Response:

The correction was made on page 10, line 376.

Line 377 and 380: “at all.” should read “et al.”

Response:

The correction was made on page 12, line 442.

Line 389: “RhoA pathway activity” should read RhoA pathway protein content”.

Response:

The correction was made on page 13, line 477–478.

Line 400: Remove the second “of”.

Response:

The correction was made on page 13, line 511.

Round 2

Reviewer 1 Report

amended manuscript is acceptable. The article is now ready for publication.

Author Response

September 08, 2022

Ms. Ivy Gong

Assistant Editor, MDPI

Ref: Biomolecules-1876278 „The protective effect of simvastatin on the systolic function of the

heart in the model of acute ischemia and reperfusion is due to inhibition of the RhoA pathway and independent of reduction of MMP-2 activity” by Skrzypiec-Spring et al.

Dear Ms. Ivy Gong,

Thank you for providing us the opportunity to further revise our manuscript. We thank the Editors and Reviewers for the time and effort that they invested into the review of our revised manuscript. We appreciate the positive comments from the Reviewers. It was your valuable and insightful comments that lead to possible improvements in the current version. 

We have implemented comments of the Reviewers and wish to submit a revised version of the manuscript for further consideration in the journal. Changes of the manuscript are marked up using the “Track Changes” function according to the instructions sent to us.

Please see below our responses to the reviewers’ comments. 

We do hope that the manuscript will now be considered suitable for publication in Biomolecules, Special Issue "Matrix Metalloproteinases in Health and Disease 3.0"

Yours sincerely,

Monika Skrzypiec-Spring MD, PhD

Department of Pharmacology, Medical University of Wroclaw 

Mikulicza-Radeckiego 2, 50-345, Wroclaw, Poland

Telephone: (+48 71) 784-14-52, Fax: (+48 71) 784-00-94

Reviewer #1 comments:

Amended manuscript is acceptable. The article is now ready for publication.

Response

We would like to thank you for your time and the appreciation of our revised manuscript. We appreciate your kindness and your help in improving the manuscript. 

Reviewer 2 Report

The authors have sufficiently expanded the results and discussion sections to reflect the data. They have also adequately identified the limitations of the study. They state that time limits prohibit further experiments. If the editor is okay with this, then that is okay with this reviewer. 

There are a few instances of "at al." and one "at. all" that should be changed to "et al.". 

Author Response

September 08, 2022

Ms. Ivy Gong

Assistant Editor, MDPI

Ref: Biomolecules-1876278 „The protective effect of simvastatin on the systolic function of the

heart in the model of acute ischemia and reperfusion is due to inhibition of the RhoA pathway and independent of reduction of MMP-2 activity” by Skrzypiec-Spring et al.

Dear Ms. Ivy Gong,

Thank you for providing us the opportunity to further revise our manuscript. We thank the Editors and Reviewers for the time and effort that they invested into the review of our revised manuscript. We appreciate the positive comments from the Reviewers. It was your valuable and insightful comments that lead to possible improvements in the current version. 

We have implemented comments of the Reviewers and wish to submit a revised version of the manuscript for further consideration in the journal. Changes of the manuscript are marked up using the “Track Changes” function according to the instructions sent to us.

Please see below our responses to the reviewers’ comments. 

We do hope that the manuscript will now be considered suitable for publication in Biomolecules, Special Issue "Matrix Metalloproteinases in Health and Disease 3.0"

Yours sincerely,

Monika Skrzypiec-Spring MD, PhD

Department of Pharmacology, Medical University of Wroclaw 

Mikulicza-Radeckiego 2, 50-345, Wroclaw, Poland

Telephone: (+48 71) 784-14-52, Fax: (+48 71) 784-00-94

Reviewer #2 comments:

The authors have sufficiently expanded the results and discussion sections to reflect the data. They have also adequately identified the limitations of the study. They state that time limits prohibit further experiments. If the editor is okay with this, then that is okay with this reviewer.

Response

We thank the Reviewer for precious time in reviewing our revised paper, the appreciation of our work and providing valuable comments. We again appreciate the very constructive comments that lead to improvements in the current version.

There are a few instances of "at al." and one "at. all" that should be changed to "et al.".

Response

We thank the reviewer for pointing out these errors. It has been corrected in the text on page 11 lines 432, 433, 435, 446 and page 12, line 452.
